Proteomic analysis of differential anther development from sterile/fertile lines in Capsicum annuum L.

Pei Hongxia 1 2
Xie Hua 2
Wang Xuemei 2
Yan Xiujuan 2
Wang Baike 3
Feng Haiping 2
Zhao Yunxia 2
Gao Jingxia 2 gjj830114@163.com
Gao Jie 1 ofc111@163.com
1 College of Horticulture, Xinjiang Agricultural University , Urumqi , China
2 Institute of Horticulture Crops, Ningxia Academy of Agriculture and Forestry Sciences , Yinchuan , China
3 Institute of Horticulture Crops, Xinjiang Academy of Agricultural Sciences , Urumqi , China
Gomez-Casati Diego
Electronic publication date: 2022 May 27
Publication date: 2022
Volume: 10
Electronic Location ID: e13168
Received 2021 Aug 30; Accepted 2022 Mar 4
Copyright: © 2022 Pei et al.
Copyright year: 2022
Copyright holder: Pei et al.
License: This is an open access article distributed under the terms of the Creative Commons Attribution License, which permits unrestricted use, distribution, reproduction and adaptation in any medium and for any purpose provided that it is properly attributed. For attribution, the original author(s), title, publication source (PeerJ) and either DOI or URL of the article must be cited.
License URL: https://creativecommons.org/licenses/by/4.0/

Keywords: Male sterility, Proteasome, Parallel reaction monitoring (PRM), Phytohormone, Pepper (Capsicum annuum L.)

Funding: Agricultural Science and Technology Innovation Fund of Ningxia NGSB-2021-8-01 National Natural Science Foundation of China 31860555 National Natural Science Foundation of Ningxia 2021A1237 The present study was funded by the Agricultural Science and Technology Innovation Fund of Ningxia (NGSB-2021-8-01), and was supported by the National Natural Science Foundation of China (31860555) and the National Natural Science Foundation of Ningxia (2021A1237). The funders had no role in study design, data collection and analysis, decision to publish, or preparation of the manuscript.

==============================
Background

Pepper (Capsicum annuum L.) is a major cash crop throughout the world. Male sterility is an important characteristic in crop species that leads to a failure to produce functional pollen, and it has crucial roles in agricultural breeding and the utilization of heterosis.

Objectives

In this study, we identified many crucial factors and important components in metabolic pathways in anther and pollen development, and elucidated the molecular mechanism related to pollen abortion in pepper.

Methods

Pepper pollen was observed at different stages to detect the characteristics associated with male sterility and fertility. The phytohormone and oxidoreductase activities were detected in spectrophotometric and redox reaction assays, respectively. Proteins were extracted from male sterile and fertile pepper lines, and identified by TMT/iTRAQ (tandem mass tags/isobaric tags for relative and absolute quantitation) and LC-MS/MS (liquid chromatograph-mass spectrometer) analysis. Differentially abundant proteins (DAPs) were analyzed based on Gene Ontology annotations and the Kyoto Encyclopedia of Genes and Genomes database according to |fold change)| > 1.3 and P value < 0.05. DAPs were quantified in the meiosis, tetrad, and binucleate stages by parallel reaction monitoring (PRM).

Results

In this study, we screened and identified one male sterile pepper line with abnormal cytological characteristics in terms of pollen development. The peroxidase and catalase enzyme activities were significantly reduced and increased, respectively, in the male sterile line compared with the male fertile line. Phytohormone analysis demonstrated that the gibberellin, jasmonic acid, and auxin contents changed by different extents in the male sterile pepper line. Proteome analysis screened 1,645 DAPs in six clusters, which were mainly associated with the chloroplast and cytoplasm based on their similar expression levels. According to proteome analysis, 45 DAPs were quantitatively identified in the meiosis, tetrad, and binucleate stages by PRM, which were related to monoterpenoid biosynthesis, and starch and sucrose metabolism pathways.

Conclusions

We screened 1,645 DAPs by proteomic analysis and 45 DAPs were related to anther and pollen development in a male sterile pepper line. In addition, the activities of peroxidase and catalase as well as the abundances of phytohormones such as gibberellin, jasmonic acid, and auxin were related to male sterility. The results obtained in this study provide insights into the molecular mechanism responsible for male sterility and fertility in pepper.

Background

Pepper is a crucial crop plant that is cultivated throughout the world. During pollen development in flowering plants, the anther structure comprising non-reproductive and reproductive tissues is enveloped by the tapetum, middle layer, endodermis, and epidermis layers from the interior to the exterior surface of pollen (Ning et al., 2019; van der Linde & Walbot, 2019). Among these structures, the tapetum acts as a nutrient source during anther development. During anther development, the microspores must acquire nutrients, sporopollenin precursors, and metabolites from the tapetum, where the inner layer of the anther wall acts as a physiological bridge between the microspores (Ariizumi & Toriyama, 2011; Blackmore et al., 2007). The sporogenous cells are surrounded by the pollen sac and they form many pollen mother cells, which then form the microspore tetrad. Finally, microspores are released from the tapetum layer to form the mature pollen grains (Ariizumi & Toriyama, 2011; Blackmore et al., 2007). All of these development processes are essential for pollen formation and development. A recent study showed that tapetal cell development, degradation, and microspore development are closely related to male fertility (Guo et al., 2018). Male sterility is characterized by abnormal male organ development and the failure to produce normal pollen grains. Male sterility in plants can be divided into genetic male sterility and cytoplasmic male sterility (Bohra et al., 2016). Cytoplasmic male sterility involves the suppression of functional pollen grain development, which is controlled by mitochondrial genes. By contrast, genetic male sterility is controlled by a pair of nuclear genes, which are stably transferred to the offspring (Kim & Kim, 2006; Yuan et al., 2018).

In genetic breeding, the male sterility system has been applied as an effective strategy to achieve heterosis in flowering plants, such as rice, cotton, and wheat (Cheng et al., 2020; Melonek et al., 2021; Tang et al., 2017). In a previous study, the rice fertility genes OsNP1 and ZmAA1 were expressed in the nuclear sterile mutant osnp1-1 to ultimately create a genetically stable sterile line (Zhen18A) and its maintainer line (Zhen18B) (Chang et al., 2016a). Recently, a deficient male-sterile mutant constructed using the CRISPR/Cas9 technique was transformed with a recombinant vector cassette containing orfH79 and CYP703A3 to obtain the maintainer line 9311-3B for heterosis breeding (Song et al., 2021). Therefore, understanding and controlling the molecular mechanisms responsible for male sterility are beneficial for plant breeders. Thus, many previous studies have aimed to identify candidate genes related to the control of cytoplasmic male sterility and the associated molecular mechanisms. For example, MutS HOMOLOG1 (MSH1) affects fertility reversion from cytoplasmic male sterility in Brassica juncea (Zhao et al., 2021b). The RFL24 protein in Arabidopsis thaliana is a mitochondrial-targeted pentatricopeptide repeat that can inhibit mitochondrial genes related to male gametes to cause cytoplasmic male sterility (Durand et al., 2021). A recent study found that Rf1 and Rf3 candidates in hybrid wheat can bind to the mitochondrial orf279 transcript and induce its cleavage to prevent the expression of cytoplasmic male sterility, thereby restoring normal pollen production in transgenic wheat plants (Geyer et al., 2016; Melonek et al., 2021). The atp6-2 gene in pepper species encodes a subunit of the mitochondrial ATP synthase complex and it may be responsible for the male sterility trait due to a deletion at the 3′-end of the male sterility allele (Arimura et al., 2020; Kim & Kim, 2006). A strong candidate gene for the male sterile, orf507, harbors a single nucleotide missing which leads the open reading frame shift to 507 bp (Gulyas et al., 2010).

The increasing availability of male-sterile genomes and transcriptome sequences has facilitated the identification of candidate factors related to cytoplasmic male sterility. In particular, transcriptome analysis of pepper 8214A and 8214B anthers detected 1,355 differentially expressed genes, including ubiquitin ligase genes, methyltransferase genes, and cell cycle-related genes (Qiu et al., 2018). In addition, transcriptional analysis of flower buds from a pepper sterile line and restorer line identified eight genes related to the phosphatidylinositol signaling system, and inositol phosphate metabolism was associated with fertility restoration (Wei et al., 2019). In pepper, 1,069 differentially expressed genes and 764 differentially expressed proteins were identified based on proteomic and transcriptomic sequences, respectively, and the TDF1 and AMS genes were associated with tapetum and pollen exine formation (Cheng et al., 2019). However, it is unclear how these candidate genes might influence male sterility. Proteomics is a powerful high-throughput method for screening candidate proteins and identifying molecular mechanisms related to male sterility pathways (Sheoran & Sawhney, 2010; Wen et al., 2007; Zhang et al., 2018). For example, 1,390 differentially abundant proteins (DAPs) associated with cytosolic ascorbate peroxidase, carbohydrate metabolism, gibberellin 20-oxidase, and enolase were identified in the nuclear male-sterile cotton line ms5ms6 (Wang et al., 2019). Thus, proteomics analysis is advantageous for exploring global protein expression and resolving the molecular mechanisms related to pollen abortion and pollen development. However, in pepper species, only a limited number of candidate proteins (23 distinct proteins) have been screened and identified by two-dimensional gel electrophoresis (Wu et al., 2013), and thus little proteomic information is available for exploring the mechanisms related to male sterility in pepper species.

In this study, we screened and identified a male sterile pepper line based on the morphological features of pollen. The peroxidase (POD) enzyme activity was significantly reduced in a male sterile line compared with a fertile pepper line, whereas the catalase (CAT) enzyme activity was significantly increased. Phytohormone analysis indicated that the gibberellin, jasmonic acid, and auxin contents differed in the male sterile pepper line compared with the fertile line. Furthermore, 1,645 DAPs were identified in the meiosis, tetrad, and binucleate stages using high-throughput proteomics. These DAPs were mainly distributed in the chloroplast and cytoplasm. In total, 45 DAPs were quantitatively identified in the meiosis, tetrad, and binucleate stages by parallel reaction monitoring (PRM), and they were related to monoterpenoid biosynthesis, and starch and sucrose metabolism pathways. Our results provide new insights into the important proteins for male sterility and the associated molecular mechanisms, thereby facilitating the cloning of genes related to male sterility.

Methods

Plant materials and growth conditions

Male fertile and sterile pepper lines were grown in a greenhouse according to standard agronomic practices. During flowering, flower buds were collected at different development stages (meiosis, microspore tetrad, binucleate, and microspore stages) and cytological observations were conducted using an Olympus CX31 microscope. In addition, three biological replicates of fertile and sterile anthers were collected and frozen rapidly in liquid nitrogen, before storing at –80 °C for proteomic analysis.

Cytological observations

Flower buds were collected and fixed in formalin:acetic acid:alcohol (ethanol:acetic acid:formaldehyde:double-distilled H2O = 10:1:2:7) for 24 h at different development stages to make cytological observations of semithin sections. The materials were then washed three times with 200 μL of 0.2 M phosphate-buffered saline and dehydrated with an ethanol concentration gradient. Finally, the samples were embedded in resin for 24 h and sectioned to obtain transverse slices using a microtome. The transverse slices were stained with 1.5–2% toluidine blue (Liu et al., 2014; Lou et al., 2007), before photographing using an Olympus CX31 microscope. In addition, pollen grains were collected from anthers using tweezers to determine their viability. The pollen grains were stained with aceto-carmine and photographed using an optical microscope.

Detection of enzyme activities and plant hormone contents

Samples of flower buds at different stages were collected and ground in liquid nitrogen to determine the oxidordeuctase enzyme activities. The ground powdered samples were extracted and incubated in extraction buffer. The homogenate was centrifuged at 8,000g and 4 °C for 15 min, and the supernatant solution was used to determine the enzyme activities with extraction kits according to the manufacturer’s instructions.

To determine the plant hormone contents, the soluble supernatant was extracted from 0.1 g of flower bud samples collected at different stages after homogenizing at 4 °C overnight in 1 mL of chilled buffer (methanol:double-distilled H2O:acetic acid = 80:20:1). The residue was extracted in chilled buffer after centrifugation at 8,000g for 10 min. The residue was centrifuged and heated up to 40 °C under a stream of nitrogen. The sample was extracted and decolorized three times in 0.5 mL of petroleum ether at 60–90 °C. Saturated citric acid aqueous solution was then added to the sample. Finally, the sample was spun to mix well after adding 0.5 mL of methanol and used for spectrophotometric assays.

Protein extraction, TMT labeling, HPLC (High Performance Liquid Chromatography) fractionation, and LC-MS/MS analysis

All samples for protein extraction were ground in liquid nitrogen and sonicated three times using an ultrasonic processor in cool lysis buffer (8 M urea and 1% protease inhibitor). Each solution was centrifuged at 12,000g and 4 °C for 10 min. The protein concentration in the supernatant was assayed using a BCA kit according to the manufacturer’s instructions. To digest the proteins, 5 mM dithiothreitol was added to the supernatant solution, which was heated for 30 min at 56 °C and then treated with 10 mM iodoacetamide to alkylate for 15 min at 56 °C in darkness. The protein solution was diluted with 100 μL of 50 mM triethylammonium bicarbonate. Finally, trypsin was added to the protein sample solution at a trypsin:protein mass ratio of 1:50 and the first protein digestion was conducted overnight at 37 °C. For the second protein digestion, the trypsin:protein mass ratio was 1:100.

For the TMT/iTRAQ assay, 0.5 M triethylammonium bicarbonate was added to the peptide solution and reconstituted using the TMT/iTRAQ kit. One unit of reagent was thawed in acetonitrile for 2 h at 25 °C. All peptides were subjected to LC-MS/MS analysis using nano-flow liquid chromatography tandem mass spectrometry (LC-MS) with an EASY-nLC 1200 UPLC system connected to a Q Exactive™ Plus quadrupole orbitrap mass spectrometer.

Protein identification and bioinformatics analysis

For protein identification, raw data were searched against Capsicum_annuum_4072 database (35,548 entries) concatenated with reverse decoy database by Proteome Discoverer search engine (V2.4.1.15). Two missing cleavages were allowed using Trypsin cleavage enzyme. The mass tolerance for precursor ions was 10 ppm and fragment ions was 0.02 Da. Carbamidomethyl on Cys, TMT-6plex (N-terminus) and TMT-6plex (K) were specified as fixed modifications. In addition, oxidation on Met and acetylation (N-terminus) were as variable modifications. The quantitative method was set as TMT-6plex. FDR was adjusted to 1% and minimum score for modified peptides was set >40. Minimum peptide length was set at 6. The raw data and results were submitted on ProteomeXchange (accession: PXD031331).

For differentially expressed proteins analysis, firstly, the expression ratio of each protein between two samples with multiple replicates were obtained and calculate the average value. Then, the significance p value of differentially expressed proteins was calculated by the two-tailed T-test method. DAPs with p value <0.05 and expression ratio >1.3 were up-regulated, while DAPs with p value <0.05 and expression ratio <1/1.3 were down-regulated proteins.

All DAPs were functionally classified into biological processes, molecular functions, and cellular components based on Gene Ontology (GO) annotations (http://www.geneontology.org/), and enriched subcategories and metabolic pathways using the Kyoto Encyclopedia of Genes and Genomes (KEGG) database (http://www.genome.jp/kegg/pathway.html). In addition, all DAPs were searched against the STRING database (V10.1) to detect protein–protein interactions (PPIs) based on confidence score >0.7. Arabidopsis thaliana was used as the reference species for establishing PPI networks. Data visualization in venn diagram were used by Tbtools software (Chen et al., 2020). The PCA diagram, volcano plot and heat map were completed by online software (https://www.ptmbiolabs.com/). The bar charts in this study were made by origin software (V.7.0).

Targeted protein quantification by PRM

Protein extraction and trypsin digestion were performed in the TMT experiment for targeted protein quantification. After LC-MS/MS analysis, the data obtained for each sample were processed using the PRM acquisition method to quantify the abundances of targeted proteins. Each protein with more than two unique peptides was quantified, but only one peptide segment was identified for some proteins due to sensitivity and other reasons. Three biological replicates of each sample were performed and 15–20 proteins were randomly selected and quantified. PRM analysis was conducted by Jingjie PTM BioLabs (Hangzhou, China) for this study.

Results

Differential anther development in male sterile and fertile pepper lines

We screened one putative male sterile pepper line that produced few pollen grains compared with the fertile pepper line. To confirm the pollen abortion characteristics in the male sterile pepper line, we observed the morphological differences in anther development during different stages using semi-thin sections. No obvious differences were found between the fertile and sterile male lines in the sporogenous cell stage and the pollen mother cell stage (Figs. 1A, 1B, 1H, and 1I). The nucleus volume was larger and the microspore mother cells entered meiosis in the male sterile line (Fig. 1C). Observations of tapetum cells indicated that they disintegrated in the male sterile line (Fig. 1J). Furthermore, the stained microspores were dispersed in the chamber in the microspore stage in the fertile pepper line (Figs. 1D–1F). By contrast, the microspores were crushed and disintegrated, and they adhered to the tapetum to form a dense belt in the male sterile line (Figs. 1K–1M). In the last stage of pollen development, normal grains were clearly observed during pollen maturation in the fertile pepper line (Fig. 1G), whereas a thin layer of tapetum cells formed a dense belt without mature pollen in the sterile line (Fig. 1N). Thus, morphological observations indicated that premature tapetum death may have been the cause of male sterility.

Figure 1 Characteristics of anthers in different development stages in male sterile and fertile lines.

(A–G) The cytological observation of anthers development stages in fertile lines; (H–N) The cytological observation of anthers development stages in male sterile lines. (A & H) The sporogenous cell stage; (B & I) the pollen mother cell stage; (C & J) meiosis stage; (D & K) the microspore tetrad stage; (E & L) the uninucleate microspore stage; (F & M) binucleate microspore stage; (G & N) pollen maturation stage. MMC, pollen mother cells; MC, meiotic cell; T, tapetum; Tds, tetrads; Msp, microspore; PG, mature pollen grain; Bar = 100 μm. (O & P) Pollen grains from male sterile and fertile lines stained by aceto-carmine. Bar = 100 μm. (Q–T) Detection of antioxidant enzyme activity in meiosis, microspore tetrad and uninucleate microspore stages. (U–X) Detection of plant hormone content in meiosis, microspore tetrad and uninucleate microspore stages.

To further assess the viability of the pollen grains, anthers were collected and the pollen grains were stained with aceto-carmine. Interestingly, large amounts of the pollen grains were strongly stained in the fertile pepper line (Fig. 1O), whereas none of the pollen grains stained in the male sterile line (Fig. 1P). These results suggest that male sterility may have been due to the production of few pollen grains and their infertility.

Antioxidant enzyme activities in different anther development stages in male sterile and fertile pepper lines

We investigated the differences in the activities of antioxidant enzymes during anther development in the male sterile (BY) and fertile (KY) pepper lines. In particular, the superoxide dismutase (SOD), CAT, POD, and malondialdehyde enzyme activity levels were compared in three anther development stages (microspore tetrad, meiosis, and uninucleate microspore stages). As shown in Figs. 1Q–1T, the POD enzyme activity levels (9,499–11,000 U/g) in the meiosis and microspore tetrad stages were significantly lower in the male sterile line compared with those in the fertile pepper line (11,000–13,800 U/g). However, the POD enzyme activity levels did not differ significantly in the male sterile and fertile pepper lines in the uninucleate microspore stage. The SOD enzyme activity levels and malondialdehyde contents did not differ significantly in the three stages in the male sterile and fertile pepper lines. In addition, the CAT enzyme activity was significantly lower in the fertile line in the three anther development stages compared with the male sterile pepper line, but especially in the uninucleate microspore stage.

Plant hormone contents in different anther development stages in male sterile and fertile pepper lines

A previous study determined significant differences in the plant endogenous hormone contents between male sterile and fertile lines in different species (Ding et al., 2018; Yang et al., 2020). Thus, in order to determine whether plant hormones are involved in differential anther development, the gibberellin, auxin, abscisic acid (ABA), and jasmonic acid contents were quantified in the male sterile and fertile pepper lines. As shown in Figs. 1U–1X, the gibberellin and jasmonic acid contents were significantly lower in the meiosis stage in the male sterile line compared with the fertile pepper line. In the microspore tetrad stage, the ABA content was slightly higher in the fertile pepper line but the same in the uninucleate microspore stage (Fig. 1W). However, the auxin content was significantly lower in the uninucleate microspore stage in the fertile line compared with the male sterile pepper line (Fig. 1V).

DAPs during anther development

To elucidate the molecular mechanisms related to male sterility, proteomic analysis was conducted to determine the differences between the male sterile and fertile pepper lines. The total proteins were extracted from the sterile and fertile lines in the meiosis, tetrad, and binucleate stages, and subjected to proteomic analysis. The proteomic analysis workflow is shown in Figs. 2A and 2B. The good repeatability of samples indicates that the result data is highly reliable (Fig. 2C). In total, 8,516 proteins and 1,646 DAPs were identified in all of the samples (Fig. 2D and Table S1). The distributions of all DAPs and their overlapping in different developmental stages were visualized by venn diagram analysis. As shown in Fig. 3, only eight DAPs (five upregulated and three downregulated) were identified in three stages (Figs. 3A and 3B). However, the number DAPs of their overlapping in two different developmental stages were no more than 50 in venn diagram (Figs. 3A and 3B). To identify proteins related to male sterility, 362 (213 upregulated and 149 downregulated), 179 (99 upregulated and 80 downregulated), and 1,104 (453 upregulated and 651 downregulated) DAPs were screened based on comparisons of BY_JS (meiosis stage in the male sterile) vs. KY_JS (meiosis stage in the male fertile), BY_SF vs. KY_SF (microspore tetrad stage), and BY_SH vs. KY_SH (binucleate microspore stage) with fold changes of 1.3 and P ≤ 0.05 (Figs. 3C–3E). The number of DAPs was significantly higher in the binucleate microspore stage than that the meiosis and microspore tetrad stages (Fig. 3E).

Figure 2 Proteome analysis of anthers at different development stages in the male sterile and fertile lines.

(A) Outline of experimental workflow. (B) Outline of proteome sequence. (C) PCA plots from proteome of anthers in meiosis, tetrad and binucleate stages in the male sterile and fertile lines. (D) Statistic of DAPs in meiosis, tetrad and binucleate stages from sterile and fertile line.

Figure 3 Differential abundance proteins (DAPs) in meiosis, tetrad and binucleate stages from sterile and fertile lines.

(A & B) Venn diagrams of DAPs between BY_JS vs. KY_JS, BY_SF vs. KY_SF and BY_SH vs. KY_SH; (C–E) Volcano plots of the DAPs in meiosis, microspore tetrad and binucleate microspore stages, repectively; JS, meiosis; SF, microspore tetrad; SH, binucleate microspore stages.

GO analysis of DAPs

To clarify the functional categories for DAPs in the male sterile and fertile pepper lines, the DAPs were assigned to three categories comprising biological process, cellular component, and molecular function (Fig. 4). Cellular process, metabolic process, and response to stimulus were the most highly enriched terms in all three stages in the biological process category, but especially in the binucleate microspore stage, and the numbers of DAPs assigned to other terms were slightly lower (Fig. 4). Interestingly, more DAPs were mainly assigned to peroxisome and microbody process in the microspore tetrad stage (Fig. 4 and Fig. S1). In the cellular component category, DAPs were significantly enriched in the cell, organelle, and membrane terms (Fig. 4). Many DAPs were assigned to translation, cytosolic ribosome process, and amide biosynthesis in the meiosis stage (Fig. S1). However, less DAPS were assigned to the extracellular region term in the meiosis stage compared with the binucleate microspore and microspore tetrad stages. Catalytic activity and binding were the most highly enriched molecular function terms in all three stages (Fig. 4).

Figure 4 Statistical distribution of DAPs under each GO annotation at meiosis, microspore tetrad and binucleate microspore stages in the male sterile and fertile lines.

(A–C) GO terms of DAPs at meiosis, microspore tetrad and binucleate stages in the male sterile and fertile lines, respectively. The x-axis represents GO terms and the y-axis represents the number of DAPs.

The functional subcategories and metabolic pathways were assigned for DAPs, and the results indicated that DAPs in the meiosis stage were mainly enriched in the protein processing in endoplasmic reticulum, proteomic regulatory subunits, and heat shock proteins pathways (Fig. 5, Table S2). In the microspore tetrad stage, the DAPs were mainly enriched in the monoterpenoid biosynthesis pathway. However, the monoterpenoid biosynthesis and starch and sucrose metabolism pathways were enriched in the binucleate microspore stage (Fig. 5).

Figure 5 KEGG pathway of DAPs at meiosis, microspore tetrad and binucleate stages between male sterile and fertile lines.

(A–C) KEGG pathway for DAPs in the meiosis, microspore tetrad and binucleate stages, respectively.

Subcellular localizations and domain enrichment analysis for DAPs

We also analyzed the subcellular compartment localizations of the DAPs, i.e., cytoplasm, chloroplast, nucleus, plasma membrane, extracellular region, mitochondrion, vacuolar membrane, and endoplasmic reticulum. As shown in Fig. 6, the DAPs in all three stages were localized in the cytoplasm, chloroplast, and nucleus, which accounted for 20–30% of the total, thereby demonstrating that proteins in these three compartments had important roles in the male sterile and fertile lines. Interestingly, the distributions of the DAPs mainly differed in the chloroplast and cytoplasm. More DAPs were localized in the cytoplasm (29%) in the meiosis stage compared with the microspore tetrad stage (24%) and binucleate microspore stage (26%) (Fig. 6A). However, more DAPs were localized in chloroplasts in the microspore tetrad stage (30%) and binucleate microspore stage (31%) compared with the meiosis stage (25%) (Figs. 6B and 6C). These different distributions indicate that many proteins in chloroplast compartments affected pollen development in the microspore tetrad and binucleate microspore stages.

Figure 6 Characteristics of DAPs in the different comparison groups.

(A) Subcellular localization of DAPs in BY_JS vs. KY_JS. (B) Subcellular localization of DAPs in BY_SF vs. KY_SF. (C) Subcellular localization of DAPs in BY_SH vs. KY_SH. (D) Enrichment of DAPs in BY_JS vs. KY_JS. (E) Enrichment of DAPs in BY_SF vs. KY_SF. (F) Enrichment of DAPs in BY_SH vs. KY_SH.

Protein domain analysis was performed at P < 0.05 using the InterPro database. In the BY_JS vs. KY_JS comparison, the enriched domains for DAPs included alpha/beta hydrolase fold, NB-ARC, and PIC domains (Fig. 6D), whereas most DAPs contained protease inhibitor and trypsin domains in BY_SF vs. KY_SF, and all were upregulated proteins (Fig. 6E). In BY_SH vs. KY_SH, the enriched domains included trypsin and protease inhibitor, PIC, and glycosyl hydrolase domains (Fig. 6F). Interestingly, the DAPs containing trypsin and protease inhibitor domains or glycosyl hydrolase domains were upregulated proteins, whereas the DAPs containing PIC domains were downregulated (Table S2).

PPI networks and cluster analysis

To understand the relationships among DAPs, we determined the PPI networks in different stages using the STRING 11.0 database. For BY_JS vs. KY_JS, 14 upregulated proteins (A0A2G2YPH5, A0A2G3AGR1, A0A1U8FF28, and others) were assigned to a ribosome network, and six upregulated proteins and one downregulated protein were clustered in a proteomic interaction network (Fig. 7A). However, only 19 DAPs comprised one small network with 13 upregulated proteins and six downregulated proteins for BY_SF vs. KY_SF (Fig. 7B). In addition, more than 100 proteins in metabolic pathways, TCA cycle (tricarboxylic acid cycle), ribosome, and basal transcription factor established a large and complex interaction network, which was closely related to plant male sterility for BY_SH vs. KY_SH (Fig. 7C). These PPI networks indicate that anther development is regulated by a complex network.

Figure 7 Protein–protein interaction network (PPI) of DAPs.

(A) BY_JS vs. KY_JS. (B) BY_SF vs. KY_SF. (C) BY_SH vs. KY_SH. The network was constructed by the String program (https://string-db.org/) with a confidence score higher than 0.4. Nodes represent proteins and the line thickness represents the strength in the picture.

Based on their similar expression patterns, all of the DAPs identified during anther development were divided into six clusters (Fig. 8). The expression levels of cluster 2 members in the sterile group (BY_JS, BY_SF, and BY_SH) were upregulated compared with those in the fertile group (KY_JS, KY_SF, and KY_SH), but the expression levels of cluster 1 members in the sterile group (BY_JS, BY_SF, and BY_SH) were downpregulated compared with those in the fertile group (KY_JS, KY_SF, and KY_SH) (Fig. 8). Furthermore, these DAPs were mainly related to citrate metabolic process, cold acclimation, regulatory region nucleic acid binding, response to jasmonic acid, microbody, and acid phosphatase activity pathway (Fig. 8 and Fig. S2). Furthermore, the expression levels in cluster 5 were significantly downregulated in KY_SH compared with BY_SH, whereas those of some in cluster 6 were upregulated in KY_SH, where they were involved in response to stress, stimulus, peroxisome, antioxidant activity, detoxification, and other metabolism pathways (Fig. 8 and Fig. S2).

Figure 8 Clustering of differential abundance proteins.

(A) DAPs identified in the anther development were divided into six clusters in BY_JS vs. KY_JS, BY_SF vs. KY_SF and BY_SH vs. KY_SH. (B) The expression patterns and different metabolic pathways of DAPs.

Targeted protein quantification by PRM

To confirm the reliability of the proteome data, PRM was conducted to quantify the targeted proteins. About 15–20 proteins from the KEGG pathway data in three different stages were randomly selected and quantified. Nineteen proteins from 20 selected randomly in the meiosis stage (BY_JS vs. KY_JS) had quantitative information and 1.5-fold changes occurred in all of these proteins, where 13 were downregulated and six were upregulated (Fig. 9A). Due to the properties of the proteins and their abundances, six proteins (four downregulated and two upregulated) were randomly selected from 15 proteins identified in the microspore tetrad stage (BY_SF vs. KY_SF) (Fig. 9B). Interestingly, three downregulated proteins with lipid-binding domains comprising A0A2G2YY91, A0A2G3AC90, and A0A1U8E7C6 may function in lipid transport during anther development. In addition, 20 proteins (10 downregulated and 10 upregulated) among 22 randomly selected proteins identified as proteins related to glycometabolism, including A0A2G2ZTG5, A0A1U8F8K6, and A0A1U8HAH9, were downregulated in the binucleate microspore stage in the male fertile pepper line (Fig. 9C). The PRM results indicate that the DAPs were stable and accurately quantified.

Figure 9 DAPs were quantified at meiosis, tetrad and binucleate stages by parallel reaction monitoring (PRM).

(A) BY_JS vs. KY_JS. (B) BY_SF vs. KY_SF. (C) BY_SH vs. KY_SH.

Discussion

Due to the higher yields of hybrid crops, heterosis has been used to breed cereals and horticultural plants, including maize, wheat, rice, and vegetable crops. It is generally considered that artificial supplementary pollination is too difficult and expensive for breeding hybrid seeds. Therefore, it is necessary to select male sterile lines to accelerate the propagation of hybrid seeds (Wang et al., 2019). In this study, we screened one male sterile pepper line and identified a set of typical features associated with male sterility, where the tapetal layer collapsed and disintegrated microspores adhered to the tapetum in a dense belt according to cytological observation. This phenomenon may be responsible for pollen abortion, as shown in previous studies (Guo et al., 2017; Han et al., 2018). However, the genes that control these traits and the molecular mechanism responsible for pollen abortion in pepper are still unclear.

Proteomic analysis is an efficient and reliable method, and it has been applied widely to understand pollen formation and development in plants. Recent studies have shown that pollen abortion in monocotyledons generally occurs in the uninucleate and binucleate microspore stages, whereas pollen abortion in dicotyledons typically occurs in the microspore tetrad stage (Guo et al., 2017; Han et al., 2018). In the present study, we conducted proteomic analysis to detect differences in DAPs in the meiosis, microspore tetrad, and binucleate microspore stages during anther development. In total, 362 (213 upregulated and 149 downregulated), 179 (99 upregulated and 80 downregulated), and 1,104 (453 upregulated and 651 downregulated) DAPs were identified in the meiosis, tetrad, and binucleate stages, respectively, which were enriched mainly in monoterpenoid biosynthesis, and starch and sucrose metabolism pathways. According to KEGG analysis, monoterpenoid biosynthesis and starch and sucrose metabolism were enriched in the binucleate microspore stage. Furthermore, glucosidase proteins related to starch and sucrose synthesis were upregulated in the fertile pepper line. Differences in the accumulation and metabolism of amino acids, monoterpenoids, and sugars are closely related to pollen abortion in fertile and sterile plant lines (Zheng et al., 2014). In the present study, many proteins related to glycometabolism were quantified as downregulated by PRM, thereby indicating that a lack of the glucose molecules required for biosynthesis and energy balance maintenance may contribute to pollen abortion in male sterile plants. Carbohydrate metabolism is one of the most basic metabolic pathways for providing energy and carbon sources in the whole plant life cycle (Wu et al., 2013). The levels of lipid transport-related proteins were also significantly reduced in the male sterile plant line, and they may play important roles in cuticle deposition on the walls of expanding epidermal cells in the tetrad stage. Exine formation is required for the absorption of essential nutrients (sugars, amino acids, and other sources) and the release of microspores from the tetrad during anther development (Chang et al., 2016b; Yang et al., 2019). Furthermore, the subcellular compartments of DAPs were identified, where most were localized in chloroplasts in the microspore tetrad and binucleate microspore stages, thereby indicating that chloroplast proteins have key roles in anther development. It is well known that carbohydrate metabolism, antioxidant reactions, photosynthesis, and plant hormone synthesis occur in chloroplasts (Chu et al., 2015; Jo, Jeong & Kang, 2009; Kurepa & Smalle, 2019). UDP-glucose 4-epimerase 3 is involved in galactose metabolism and it was downregulated in the male sterile line, thereby suggesting that the tricarboxylic acid cycle, photosynthesis, and glycometabolism were less active, thereby contributing to male sterility. These new candidates related to male sterility may facilitate the cloning of key factors associated with male sterility in future research.

Male sterility is also associated with energy metabolism disorders, the accumulation of reactive oxygen species (ROS), programmed cell death, and glycolysis in male sterile wheat according to proteomic analysis of pollen (Zhang et al., 2016). Previous studies have demonstrated that programmed cell death and ROS stress occur in the tapetum to affect the provision of nutrients and metabolites for microspore development in male sterility (Ariizumi & Toriyama, 2007; Guo et al., 2017). The tapetal layer is essential for pollen grain development because it is the site of synthesis for proteins, which are then transported to the anther walls (Qiu et al., 2016; Yang et al., 2014). Male sterility in Arabidopsis thaliana is associated with microspore development due to premature or delayed tapetal programmed cell death (Ariizumi & Toriyama, 2007). Our morphological observations indicated the occurrence of premature death and vacuolation with the crushing of the microspores in the tapetal stage in the male sterile line, thereby leading to callosum degradation and microspore release in the uninucleate stage (Fig. 2). The dynamic balance between enzymatic and nonenzymatic systems in plant cells allows antioxidant enzymes such as POD, CAT, SOD, and ascorbate peroxidase to eliminate ROS. In this study, we detected the activities of antioxidant enzymes (SOD, POD, and CAT) and found that the activities of all were significantly lower in the male sterile line than the fertile line, thereby indicating that excess ROS could not be removed in the meiosis and microspore tetrad stages. Excess ROS leads to programmed cell death and peroxidation of plant membrane lipids (Geng et al., 2018). We suggest that ROS may trigger programmed cell death and lead to male sterility in the pepper line. In addition, phytohormones are now recognized as vital factors that regulate plant development and growth, reproductive processes, biotic and abiotic stress responses, and tissue senescence (Bleecker & Kende, 2000; Zhao et al., 2021a). Previous studies have identified the specific roles of plant endogenous hormones in male sterility and fertility (Dubas et al., 2013; Shukla & Sawhney, 1994). Therefore, we assessed phytohormones in the male sterile and fertile lines to detect any differences. We found significant differences in the endogenous ABA, auxin, and gibberellin contents in different bud development stages. The ABA concentration was higher in younger floral buds. The different ABA contents may affect plant pollen abortion by specifically blocking the transport of apoplastic sugars and other carbohydrates during pollen development (Ding et al., 2018). The tomato male sterile line jai1-1 is a mutant with defective jasmonic acid perception, where SlMYB21 positively regulates jasmonic acid biosynthesis but negatively regulates auxin and gibberellin (Schubert et al., 2019). The overexpression of AtMYB2, which is the target of JAZ repressors, can also partially rescue male fertility in Arabidopsis stamen and pollen development (Song et al., 2011). In general, phytohormones and antioxidant enzymes are key factors that regulate male sterility in pollen development.

Conclusions

In this study, we employed cytological analysis, bioinformatics, proteomics, and plant physiological assays to gain insights into the mechanisms associated with male sterility in pepper. We screened 1,645 DAPs from a male sterile pepper line based on proteome analyses and 45 DAPs were quantitatively identified in the meiosis, tetrad, and binucleate stages by PRM, which were related to monoterpenoid biosynthesis, and starch and sucrose metabolism pathways. The DAPs were mainly distributed in the chloroplast and cytoplasm. In addition, the POD and CAT enzyme activities were significantly lower in the male sterile line compared with the fertile line, and the gibberellin, jasmonic acid, and auxin contents also differed. The results obtained in this study improve our understanding of the essential factors related to pollen development in male sterility.

Supplemental Information

Supplemental Information 1 Gene ontology enrichment of DAPs in three categories.

A: BY_JS vs. KY_JS. B: BY_SF vs. KY_SF. C: BY_SH vs. KY_SH. BP, biological process; MF, molecular function; CC, cellular component.

Click here for additional data file.

Supplemental Information 2 Gene ontology enrichment of DAPs from six gene clusters.

A: Gene ontology enrichment of DAPs in biological process; A: Gene ontology enrichment of DAPs in cellular component; C: Gene ontology enrichment of DAPs in molecular function. BP, biological process; MF, molecular function; CC, cellular component.

Click here for additional data file.

Supplemental Information 3 The total proteins were extracted from the sterile and fertile lines in the meiosis, tetrad and binucleate stages, and 8,516 proteins were identified in all samples.

Click here for additional data file.

Supplemental Information 4 The functional subcategories and metabolic pathways were assigned for DAPs.

DAPs in the meiosis stage were mainly enriched in the protein processing in endoplasmic reticulum, proteomic regulatory subunits, and heat shock proteins pathways.

Click here for additional data file.

Additional Information and Declarations

Competing Interests

Author Contributions

Data Availability

The authors declare that they have no competing interests.

Hongxia Pei conceived and designed the experiments, performed the experiments, analyzed the data, prepared figures and/or tables, authored or reviewed drafts of the paper, and approved the final draft.

Hua Xie performed the experiments, analyzed the data, authored or reviewed drafts of the paper, and approved the final draft.

Xuemei Wang performed the experiments, authored or reviewed drafts of the paper, and approved the final draft.

Xiujuan Yan performed the experiments, authored or reviewed drafts of the paper, and approved the final draft.

Baike Wang performed the experiments, prepared figures and/or tables, and approved the final draft.

Haiping Feng performed the experiments, prepared figures and/or tables, and approved the final draft.

Yunxia Zhao analyzed the data, prepared figures and/or tables, and approved the final draft.

Jingxia Gao conceived and designed the experiments, authored or reviewed drafts of the paper, and approved the final draft.

Jie Gao conceived and designed the experiments, authored or reviewed drafts of the paper, and approved the final draft.

The following information was supplied regarding data availability:

The raw measurements are available in the Supplemental Files.

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
