# Peer review of "Proteomic analysis of differential anther development from sterile/fertile lines in Capsicum annuum L"

_PeerJ, doi:10.7717/peerj.13168_

## Round 0.1 · original submission · Major Revisions

Please, find below the reviewer´s suggestions.

Reviewer 1 ·

Basic reporting

In this manuscript by Pei et al., the authors perform proteomic analysis of male sterility and fertile pepper lines. The data presented could be potentially valuable. There are some issues which need to be clarified:
1, Abstract in Page 4 and Page 7 is different, please check it.
2, I have noticed that some abbreviations were introduced at the beginning of this manuscript, while it was appeared the full name again at the end, such as the DAP, KEGG etc.
3, Keywords: “Proteasome” should change to “Proteomic”.
4, The English must be deeply edited,especially some long sentence, which makes reader difficult to understand. As it is presented, the manuscript is far from the standards of the scientific (biochemical) language. Besides that, the format in this manuscript also needs to be carefully modified, such as line 74, line 86 ,line 93 and so on.
5, The literature on proteomics of male sterility in pepper is not fully described in the background part, for example Cheng et al 2019 ijms.
6, Figure 1, It is necessary to indicate what period the letters in the figure represent.
7, Are buds or anthers taken for proteomic analysis? From Figure 2, it seems that anthers are taken.
8, The format of references should be uniform, for example, Latin names, gene name and mutant name need to be italicized (line 537, 500, 546 etc.); the first letter of the quoted title should accordance, most of them are lower case, while line470, 510 and so on are capital; line 470 have no volume or page number. Line 568: reference title is wrong,please check it. The name of the reference journal need to be unified, whether to use abbreviation or full name.

Experimental design

no comment

Validity of the findings

no comment

Additional comments

no comment

Reviewer 2 ·

Basic reporting

The paper needs to be proof read for grammar and spellings. I found several instances and few of them are mentioned below:

1) Line 34: "diffenrent"; Line 39: "Encyclopaedia"; Line 52-53: "This study largely improves our understanding of key fators through proteome analyses and revealed components of metabolic pathway impacted the anther and pollen development in the male sterility". This line does not make sense grammatically.

2) Line 66: "The sporogenous cells were surrounded by" this switches the paragraph from present to past. Please check this.

3) Line 44: The sentence starts with this "cytoplasmic male". It should be "Cytoplasmic male...".

Experimental design

All bioinformatics methods seemed to be thoroughly performed and documented.

Just one minor question: Why was a fold change of 1.3 used to detect differentially abundant proteins? I have seen people using a cut-off of 0 or 1 or 1.5 but never seen anybody use 1.3 before. Was there a rational behind using such an arbitrary cut-off?

Validity of the findings

The authors seemed to have applied appropriate bioinformatics tools and techniques to determine the genes and pathways responsible for male sterility in Capsicum annuum L. The results seem to be compelling and statistically reliable.

Additional comments

Please double check all the figures for clarity, a few figures are not clear and appear to be pixelated.

---

## Round 0.2 · Minor Revisions

The new version of the manuscript was reviewed by one of the reviewers and by myself. In this version, the authors took into account the suggestions of the reviewers and corrected the text appropriately. However, there are some points to clarify:

The methods need some more detail. How are the figures made? Why was a PCA performed? It isn't listed in the methods or results anywhere, just appears as a figure. Same for the Venn diagrams - how were they made, why do them, etc.

Reviewer 2 ·

Basic reporting

All issues seem to have been addressed well. Thanks!

Experimental design

N/A

Validity of the findings

N/A

Additional comments

N/A

---

## Round 0.3 · Minor Revisions

Authors made almost all the suggested changes. However, there's is no information on the criteria for accepting protein identification. Peptide and protein identification scores need to be validated, e.g. using a Target-Decoy strategy, and post-processing results, e.g. with PeptidePropet/ProteinProphet. In addition, raw data (MS spectra in mzML format) and results should be made available, e.g. on ProteomeXchange/Pride or Zenodo.

---

## Round 0.4 · Minor Revisions

The comments were partially resolved. However, the level of information is still insufficient for publishing proteomics data:

1) The manuscript states
"Protein identification and bioinformatics analysis
The LC/MS data were processed using the Capsicum_annuum_4072 (35548 sequences) database according to a peptide tolerance of 10.0 ppm and MS/MS tolerance of 0.02 Da. Two amino acids were cleaved by trypsin and the fixed modifications comprised carbamido methylation of cysteine. Proteins were identified according to the number of unique peptide ≥ 1. DAPs were selected according to |(fold change)| > 1.3 and P value < 0.05. "

Which software was used for data processing, and which tools and algorithms to validate peptides/proteins and estimate probabilities?

2) The Proteomics data are not accessible:

From the webpage search:

"ProteomeXchange dataset PXD031331 has been reserved by the PRIDE repository for a dataset that has been deposited, but is not yet publicly released and announced to ProteomeXchange.

---

## Round 0.5 · accepted · Accept

The authors corrected the text and have added all the suggestions required by the reviewers.